# Urban pollution greatly enhances formation of natural aerosols over the Amazon rainforest

Manish Shrivastava [1], Meinrat O. Andreae [2,3,4], Paulo Artaxo [5], Henrique M.J. Barbosa [5], Larry K. Berg [1], Joel Brito [6], Joseph Ching [7], Richard C. Easter[1], Jiwen Fan [1], Jerome D. Fast [1], Zhe Feng [1], Jose D. Fuentes [8], Marianne Glasius [9], Allen H. Goldstein [10], Eliane Gomes Alves [11], Helber Gomes [12], Dasa Gu[13], Alex Guenther[1,13], Shantanu H. Jathar [14], Saewung Kim[13], Ying Liu [1], Sijia Lou [1], Scot T. Martin[15], V. Faye McNeill[16], Adan Medeiros[17], Suzane S. de Sá[15], John E. Shilling [1], Stephen R. Springston [18], R.A.F. Souza [19], Joel A. Thornton[20], Gabriel Isaacman-VanWertz [21], Lindsay D. Yee [10], Rita Ynoue [22], Rahul A. Zaveri [1], Alla Zelenyuk [1] & Chun Zhao[23]

One of the least understood aspects in atmospheric chemistry is how urban emissions influence the formation of natural organic aerosols, which affect Earth's energy budget. The Amazon rainforest, during its wet season, is one of the few remaining places on Earth where atmospheric chemistry transitions between preindustrial and urban-influenced conditions. Here, we integrate insights from several laboratory measurements and simulate the formation of secondary organic aerosols (SOA) in the Amazon using a high-resolution chemical transport model. Simulations show that emissions of nitrogen-oxides from Manaus, a city of ~2 million people, greatly enhance production of biogenic SOA by 60–200% on average with peak enhancements of 400%, through the increased oxidation of gas-phase organic carbon emitted by the forests. Simulated enhancements agree with aircraft measurements, and are much larger than those reported over other locations. The implication is that increasing anthropogenic emissions in the future might substantially enhance biogenic SOA in pristine locations like the Amazon.

[1] Pacific Northwest National Laboratory, Richland, WA 99352, USA. [2] Department of Geology and Geophysics, King Saud University, Riyadh 11451, Saudi Arabia. [3] Scripps Institution of Oceanography, University of California San Diego, La Jolla, CA 92093-0230, USA. [4] Max Planck Institute for Chemistry, P.O. Box 3060, Mainz D-55020, Germany. [5] Institute of Physics, University of São Paulo, São Paulo 05508-090, Brazil. [6] IMT Lille Douai, University of Lille, SAGE, Lille 59000, France. [7] Meteorological Research Institute, Japan Meteorological Agency, 1-1, Nagamine, Tsukuba, 305-0052 Ibaraki, Japan. [8] Department of Meteorology and Atmospheric Science, Penn State University, University Park, PA 16802, USA. [9] Department of Chemistry, Aarhus University, Aarhus 8000, Denmark. [10] Department of Environmental Science, Policy, and Management, University of California, Berkeley 94720, USA. [11] Instituto Nacional de Pesquisas da Amazônia (INPA), Av. André Araújo, Manaus, AM 69.060-000, Brazil. [12] Institute of Atmospheric Sciences, Federal University of Alagoas, Maceió, AL 57072-900, Brazil. [13] Department of Earth System Science, University of California, Irvine, CA 92697, USA. [14] Department of Mechanical Engineering, Colorado State University, Fort Collins 80523, USA. [15] School of Engineering and Applied Sciences and Department of Earth and Planetary Sciences, Harvard University, Cambridge, MA 02138, USA. [16] Department of Chemical Engineering, Columbia University, New York, NY 10027, USA. [17] Amazonas State University, Center of Superior Studies of Tefé, R. Brasília, Tefé, AM 69470000, Brazil. [18] Environmental and Climate Sciences Department, Brookhaven National Laboratory, Brookhaven, NY 11973, USA. [19] Amazonas State University, Superior School of Technology, Av Darcy Vargas, Manaus, AM 69050020, Brazil. [20] Department of Atmospheric Sciences, University of Washington, Seattle 98195, USA. [21] Department of Civil and Environmental Engineering, Virginia Tech, Blacksburg, VA 24061, USA. [22] Department of Atmospheric Sciences, Institute of Astronomy, Geophysics and Atmospheric Sciences, University of Sao Paulo, Sao Paulo 05508090, Brazil. [23] School of Earth and Space Sciences, University of Science and Technology of China, Hefei 230026, China. Correspondence and requests for materials should be addressed to M.S. (email: ManishKumar.Shrivastava@pnnl.gov)

The response of natural systems to anthropogenic emissions remains one of the largest uncertainties in our understanding of the radiative forcing of climate[1–3]. Secondary organic aerosol (SOA) is a ubiquitous component of atmospheric aerosol, which scatters and absorbs solar radiation and also activates to form cloud droplets[4–6]. Over pristine regions such as the Amazon rainforests, SOA formed by oxidation of biogenic volatile organic compound (VOC) precursors accounts for most of the cloud condensation nuclei, especially during the wet season[7]. Field measurements suggest that much of biogenic SOA mass is formed through mechanisms that are driven/enhanced by anthropogenic emissions[8–12]. Anthropogenically controlled biogenic SOA refers to SOA formed due to oxidation of biogenic precursors, but that would not be formed in absence of anthropogenic emissions[13]. A modeling study over the United States suggested that ~20% of biogenic SOA was controlled by anthropogenic nitrogen-oxides ($NO_x$) and another 30% was controlled by partitioning of SOA within primary organic aerosol (POA)[9]. Another global modeling study suggested that addition of large amounts of SOA sources (70% of total) that spatially matched anthropogenic pollution was needed to produce best model-measurement agreement[14].

Chemical pathways of SOA formation could be broadly classified as two types: [1] Pure gas-phase chemistry, which refers to gas-phase oxidation of volatile organic compounds (VOCs) emitted from terrestrial vegetation and combustion activities (e.g., wildfires, traffic) that results in formation of lower volatility condensable products[15–17] and has been studied in outdoor chambers as early as 1982[18], and [2] Multiphase chemistry, which refers to chemistry occurring between gas- and particle-phases, such as acid-catalyzed reactive uptake of organics in the aqueous phase of hygroscopic particles (aqueous aerosols)[19–21]. A significantly improved understanding of pathway 2 has developed only recently over the past decade, e.g. the uptake of isoprene epoxydiols (IEPOX) on aqueous aerosols mediated by $SO_2$ and $NO_x$[22–29]. IEPOX-SOA constitutes 10–30% of SOA at various locations around the globe[25]. However, other pathways for anthropogenic-biogenic interactions may be important as well, e.g. nonlinear effects of $NO_x$ on both gas- and particle-phase chemistry of SOA, as discussed in a recent review article[30].

One of the challenges in accurately quantifying anthropogenically controlled or enhanced biogenic SOA through field measurements is the need to establish a baseline biogenic SOA level that would exist in absence of any anthropogenic perturbations. This is difficult in large part due to the ubiquitous influence of anthropogenic emissions over most terrestrial locations in the Northern hemisphere including the continental United States. The vast Amazon rainforest during its wet season is one of the few remaining places on Earth where atmospheric chemistry transitions between pristine-preindustrial and urban pollution-influenced conditions. This region presents a unique natural laboratory to understand how anthropogenic emissions impact biogenic SOA formation[31].

While there are several observational studies during the Green Ocean Amazon (GoAmazon2014/5) field campaign over the Amazon rainforest[31], we present the first dedicated modeling study of SOA formation investigating this field campaign. We include SOA chemistry pathways observed in several laboratory studies within a high-resolution regional chemical transport model, and provide a holistic view of how natural biogenic SOA formation changes due to its chemical interactions with urban pollution. Manaus, a city of 2 million people located within the forest, represents the only major anthropogenic source within the Amazon during the wet season. In the absence of Manaus emissions, the Amazon atmosphere in the wet season approaches preindustrial conditions[7]. Measurements demonstrate a sharp contrast in levels of various pollutants (including gases, particles, and oxidants) between air masses in the pristine Amazon, and when Manaus emissions mix with this pristine air[32].

Observational studies over the Amazon have demonstrated complex interactions between urban pollution and biogenic SOA. By using an oxidation flow reactor, Palm et al.[33] showed that additional SOA could be produced from biogenic precursors (available in the ambient air) provided that additional ozone and OH concentrations were available. An analysis of VOC concentrations and variation with $NO_y$ by Liu et al.[34] indicated a substantial increase in oxidant concentrations in the pollution plume, which, considering the work of Palm et al.[33], suggests a significant role for anthropogenic control on SOA production. Statistical and cluster analysis of Aerosol Mass Spectrometer (AMS) and auxiliary datasets by de Sá et al.[35] at a surface site showed that the increase in OA ranged from 25% to 200% under polluted compared to background conditions, including contributions from both primary and secondary particulate matter. de Sá et al.[23] further showed that the relative contribution of the IEPOX-SOA factor to OA decreased significantly under polluted conditions. Liu et al.[36] observed that the afternoon concentrations of organic hydroxyhydroperoxides (ISOPOOH) decreased from 600 pptv under background conditions to <60 pptv under polluted conditions, suggesting important shifts in the gas-phase chemistry that could affect OA production. Aircraft measurements from the Manaus plume on the same day targeted in this study (March 13) found that the composition of the downwind OA became progressively more oxidized, with a conversion from hydrocarbon-like OA to oxygenated OA[37]. Our present modeling study aims to provide a mechanistic understanding of observed impacts of anthropogenic emissions on SOA formation over the Amazon.

Using a high-resolution regional chemical transport model, we contrast SOA production in air masses from the near-pristine background with those affected by the Manaus plume, in order to understand how anthropogenic emissions affect biogenic SOA formation in this region. Model predictions are evaluated with aircraft measurements of organic aerosols (OA, which is sum of POA and SOA) using a high-resolution Aerosol Mass Spectrometer (AMS)[37]. Our study focuses on aircraft measurements since the aircraft rapidly measures trace gases and aerosol concentrations over both background and plume-affected locations, concomitantly. Aircraft measurements, thus represent a snapshot of changes that occur in biogenic SOA due to anthropogenic emissions over the otherwise pristine wet-season Amazon. Most of the analyses presented in this Manuscript are for 13 March 2014, a day of mostly sunny skies and no precipitation along the aircraft flight path, which is ideal for studying SOA formation[32].

We show large enhancements (60–200% on average, 400% maximum) of natural biogenic SOA within the Amazon that are due to substantial increase in oxidants (OH and ozone) promoted by $NO_x$ emissions within the urban plume, and are much larger than the enhancements reported in other locations. In the absence of the urban plume, background $NO_x$ concentrations are much lower (that can mainly to be attributed to soil $NO_x$ emissions) causing lower OH and ozone production, thus decreasing reacted biogenic VOCs and SOA formation. We show that although isoprene dominates the emissions fluxes of biogenic VOCs within the Amazon, it contributes 50% to biogenic SOA formation while terpenes contribute the remaining half. Our results provide a clear mechanistic picture of how anthropogenic emissions are likely to have greatly enhanced biogenic SOA formation since preindustrial times over the Earth.

## Results

**Simulating OA within a regional model.** Comparing modeled and observed particle concentrations over the Amazon is particularly challenging due to large uncertainties in emissions of biogenic VOCs and a complex wet scavenging environment[38]. We use the regional Weather Research and Forecasting Model coupled to chemistry (WRF-Chem) model[39,40] at high resolution with 2 km grid spacing i.e. at cloud-, chemistry-, and emissions-resolving scales to simulate atmospheric chemistry and SOA formation during GoAmazon2014/5 (Methods). We simulate the atmospheric conditions between 10 and 17 March 2014 with the first 3 days used for spin-up of aerosol and trace gas concentrations, for a region that includes the Amazon basin (Methods, Supplementary Figure 1 and Supplementary Table 1). Due to the large computational costs associated with our SOA parameterizations and high-resolution coupled cloud-chemistry-meteorological WRF-Chem simulations (Methods), we only conduct simulations for a 1-week period. However, the results and conclusions from this study are expected to apply more broadly over the entire wet season period, since both observations and a previous WRF-Chem study show that the sharp contrast between plume and background oxidants is a common feature among several days[41].

Simulated SOA from pure gas-phase chemistry pathway is represented in the model using a modified volatility basis set (VBS) approach (Methods). The VBS approach represents multiple generations of oxidation of biogenic VOCs that include isoprene, monoterpene, and sesquiterpene compound classes, and anthropogenic and biomass burning precursors using a lumped set of compounds. Initial yields are determined by fitting environmental chamber measurements and generally vary with VOC, $NO_x$, and oxidants (Supplementary Table 2). This work also includes several major updates to SOA aging parameterizations (Methods) to gain insights into biogenic SOA formation and its interactions with anthropogenic emissions. Isoprene SOA is formed in the model by two different pathways: gas-phase chemistry pathway (represented by VBS) and multiphase IEPOX-SOA pathway (represented by simple Gamma model)[42], which are coupled to the Model for Simulating Aerosol Interaction and Chemistry (MOSAIC) aerosol module[43] within WRF-Chem (Methods).

**Aircraft measurements of OA and model predictions.** To understand how the Manaus plume affects biogenic SOA formation, we compare results from two model simulations: Default, wherein all emissions including those from Manaus urban region and biogenic emissions are on, and a sensitivity simulation wherein biogenic emissions from the forest are on, but anthropogenic Manaus emissions are turned off (including $NO_x$, $SO_2$, anthropogenic VOCs, and primary particulate emissions, e.g., POA, sulfate). This simulation represents background concentrations of OA and trace gases over the Amazon.

Figure 1a compares measured variations of OA mass concentrations using the Aerosol Mass Spectrometer (AMS)[37] with WRF-Chem predictions along aircraft flight transects on March 13. The periodic rise and fall of OA in Fig. 1a reflects times when the aircraft intersected the plume in transverse patterns (Supplementary Figure 2), thus concomitantly sampling air masses both in-plume and within the local background. Transect 1 is closest to the city (24 km downwind of T1, urban center), while transects 2, 3, and 4 are farther downwind of the city, spaced approximately equally at intervals of 24 km (Supplementary Figure 2). On this day, the default model (blue lines) also predicts that the plume emitted from the Manaus region (close to the T1 site) passed over the T3 site, which is 70 km downwind of the urban region (Figs. 2 and 3). Both measurements (orange line) and model predictions (blue line) show that OA concentrations are clearly enhanced in the plume compared to the background. Similarly, $NO_y$, ozone, and CO concentrations show a periodic rise and fall as the aircraft moves within and outside of the urban plume (Supplementary Figure 3a, 3b, and 3c, respectively). Simulations (blue line) agree with measurements (orange line) and capture this rise and fall of OA, $NO_y$, CO, and ozone corresponding to the four different aircraft transects.

The model predicts that natural biogenic SOA formed by oxidation of biogenic VOC emissions (including isoprene, monoterpenes, and sesquiterpenes, emitted by the forest, denoted by pink shaded region in Fig. 1a) dominates over anthropogenic OA (black shaded region in Fig. 1a). Anthropogenic OA is significant only within plume. The default model (blue) agrees with measured OA loadings (orange) during three of the four flight transects, however, the sensitivity simulation with Manaus emissions turned off (green) predicts much lower OA loadings,

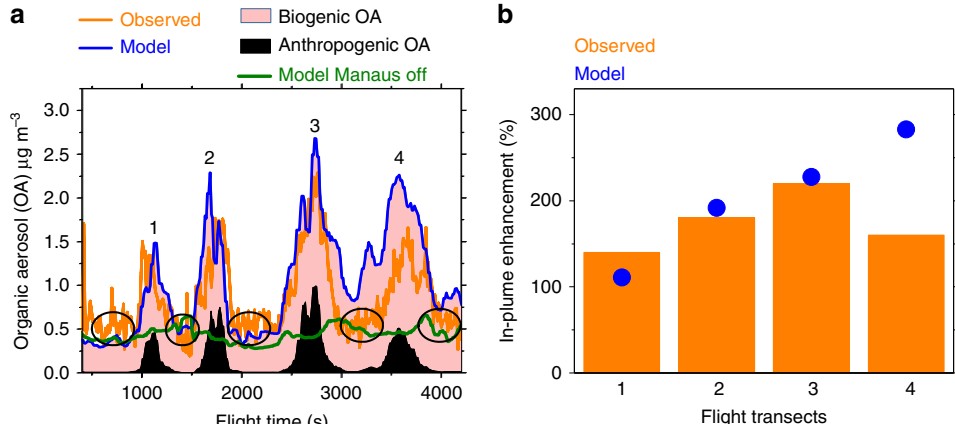

**Fig. 1** Aircraft measurements and model predictions demonstrating large OA enhancements within the urban plume compared to the pristine Amazonian background. **a** Total organic aerosols (OA) measured along aircraft flight transects at 500-m altitude on March 13 (orange) and model-predicted OA for simulations with all emissions on (blue), and biogenic emissions on but Manaus emissions off (green, representing background OA), as described in the text. The shaded regions depict simulated biogenic and anthropogenic OA components. **b** Measured and model predicted average percent enhancement in plume compared to background organic aerosol on 4 different flight transects, as marked in **a**. The figure shows four different transects when the aircraft intersected the Manaus plume marked as 1–4, while background (outside plume) concentrations are ~0.5 µg m⁻³, indicated by the open circles. Bars represent measurements while symbols represent model-predicted increases of OA within Manaus plume compared to background conditions

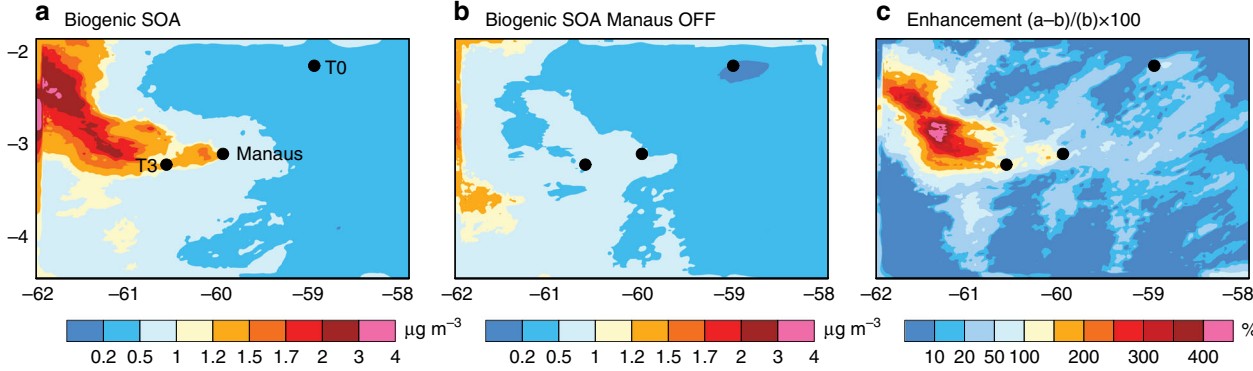

**Fig. 2** WRF-Chem simulated concentrations of biogenic SOA in the presence and absence of Manaus emissions. **a** Biogenic SOA when all emissions are on **b** Biogenic SOA when biogenic volatile organic compound (VOC) emissions are on but Manaus (anthropogenic) emissions are turned off **c** Biogenic SOA enhancement (%) calculated from the two simulations with Manaus emissions turned on/off i.e. (a–b)/b × 100. WRF-Chem predictions are at ~500 m altitude, averaged during the afternoon (16–20 UTC = 12–16 local time) of 13 March 2014

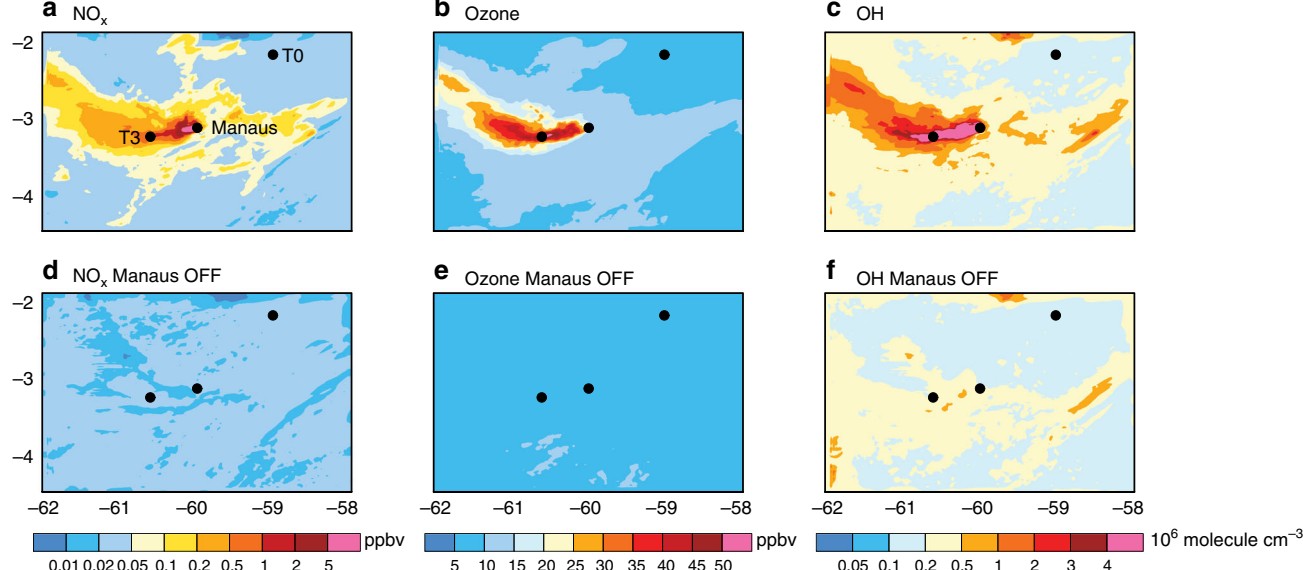

**Fig. 3** WRF-Chem simulated concentrations of $NO_x$ and oxidants with Manaus emissions turned on/off. **a**, **b**, and **c** show simulated $NO_x$, ozone and OH with all emissions on, while **d**, **e** and **f** show simulations with biogenic emissions on but Manaus emissions turned off for an altitude of ~500 m, averaged during the afternoon (16–20 UTC) of 13 March 2014. A comparison of top and bottom panels demonstrates how $NO_x$ and oxidants are greatly enhanced by the Manaus plume within the otherwise pristine Amazon

representing the regional background biogenic SOA concentrations (~0.5 µg m⁻³). The simulated background OA agrees with aircraft measurements (Fig. 1a and Supplementary Table 3). Differences between our two model simulations represented by the blue and green lines in Fig. 1a represents the enhancement of OA due to anthropogenic emissions.

In-plume model simulated OA agrees with measured OA (within 15%) for transects 1, 2, and 3. However, for transect 4, the model overestimates OA by ~50% compared to measurements. Since transect 4 is farthest from the city center, differences between locations of the observed and simulated plume are greatest for transect 4 compared to transects 1, 2, and 3, as observed from CO comparisons for this day (Supplementary Figure 3c). Consistently, Fig. 1a also shows that simulated OA (blue line) drops to the background value of ~0.5 µg m⁻³ when the aircraft moves out of the plume during flight transects 1–3 (open circles in Fig. 1a). However, simulated OA does not drop to the background value when the aircraft moves out of plume during transect 4 due to model-measurement differences in

plume location and dispersion (e.g. at ~4000 s flight time in Fig. 1a). These differences in plume location and dispersion farther downwind of Manaus partly explain the increased model-measurement differences in simulated OA for transect 4 compared to the other transects (Supplementary Table 3). Therefore, our approach focuses on identifying the shifted urban plume in the model and contrasting plume-to-background concentrations, as described below. Traditional statistical techniques for model evaluation would not be expected to work well in this study because shifts in plume location could be an issue, especially in high-resolution model simulations.

To quantify the plume-to-background enhancement in OA concentrations, we classify in-plume and background locations in measurements based on $NO_y$ thresholds (background: $NO_y < 0.5$ ppb, in-plume: $NO_y > 2$ ppb)[36]. In model simulations, we define the plume based on $NO_y$ threshold ($NO_y > 2$ ppb), similar to measurements. Background concentrations in the model are obtained from our sensitivity simulation where anthropogenic (Manaus) emissions are turned off. Both measurements and

model simulations show that $NO_y$ increases from sub-ppb levels in the background to several ppbs in-plume i.e. more than an order of magnitude increase (Supplementary Figure 3a). Since the simulated plume could be shifted compared to measurements, we determine in-plume locations in the model by scanning all radial grid-points downwind of Manaus (T1 site) that exceed the $NO_y$ threshold of 2 ppb. The model is sampled at the same time, altitude, and radial distance as corresponding aircraft measurements. Percentage enhancement factors in the model are calculated as the ratio of difference between plume and background OA (i.e. plume OA-background OA) to background OA concentrations for all locations averaged across each flight transect, similar to measurements. Percentage in-plume enhancement factors of OA compared to its immediate background are calculated individually for four different flight transects of the plume (Fig. 1b). Measurements (orange bars in Fig. 1b) indicate that OA is enhanced by an average of 100–200% in-plume with peak enhancements (calculated as the difference between largest in-plume OA concentrations to background levels) of ~400% compared to background on March 13. These enhancements are much larger than those reported in previous studies over other regions in the Northern hemisphere[9,14]. We attribute these large enhancements mostly to the sharp increase in oxidants within Manaus plume compared to the background Amazon, as discussed later. The model shows excellent agreement with observed enhancements for transects 1, 2, and 3 (within 20%) (Fig. 1b). For transect 4, the model overestimates enhancement by ~75% compared to observations due to overestimation of simulated in-plume OA for this transect.

Although the focus of this study is on March 13, which represented a golden day due to sunny conditions and clear evolution of the plume downwind of Manaus, OA was also enhanced on other days. For example, we calculated enhancements of OA in plume-affected locations on 2 other days i.e. March 14 and 16 during the simulated period based on measured and simulated $CO/NO_y$ (Supplementary Figure 4). The model moderately overestimates OA enhancements on both days (by ~50–60%). Some of these differences between simulated and measured enhancements are due to differences in plume location and dispersion.

Our simulations indicate that biogenic SOA is the dominant contributor to total OA downwind of the city (Fig. 1a and Supplementary Figure 5). This result explains the dominant oxygenated organic aerosol (OOA) contribution to total OA downwind, suggested by AMS factor analysis[37]. Consistent with our findings of OA enhancement from aircraft, a recent study also found 25–200% enhancement in submicron particles observed over the T3 ground site during the entire wet season[35].

**Simulated enhancement in biogenic SOA**. Isoprene and other biogenic VOCs are emitted throughout the Amazonian rainforest as diffuse area sources. Manaus emissions interact with these biogenic sources, increasing oxidants and SOA formation. WRF-Chem simulated spatial distribution of total biogenic SOA (sum of SOA formed by oxidation of isoprene, monoterpenes, and sesquiterpenes) from the default (all emissions on) and the simulation with Manaus emissions off are shown in Fig. 2a, b, respectively. Biogenic SOA formation is enhanced both in-plume and its outflow regions by a factor of 100–400% on average during the afternoons of 13 March 2014, as indicated in Fig. 2c, consistent with the enhancement in total OA shown for flight transects in Fig. 1.

**Background versus in-plume oxidants**. In the wet season, locations not affected by Manaus can approach conditions characteristic of preindustrial times. WRF-Chem predicts that background oxidants are mainly sustained by catalytic effects of natural NO emissions (soil $NO_x$, described below) on OH concentrations through reactions of NO with hydroperoxyl radicals ($HO_2$) and organic peroxy radicals ($RO_2$) during the daytime, and this chemistry also affects ozone. Additional OH recycling mechanisms have been suggested in the literature[44]; however, these recycling mechanisms often cause substantial overestimation of observed OH[45]. Therefore, no additional OH recycling mechanisms are included in the model.

**Soil $NO_x$ as the driver of background oxidizing capacity**. The dominant natural background source of $NO_x$ is emissions from soils, which we include here as an effective soil NO emissions flux of $8.3 \times 10^9$ molecules $cm^{-2} s^{-1}$ within WRF-Chem (Methods). This value is close to the soil $NO_x$ emissions range suggested by field measurements over Amazon rainforests, as discussed by Liu et al.[36]. Under background Amazonian conditions, the relative reaction rate of isoprene peroxy radicals (ISOPOO) with NO to that with $HO_2$ is suggested by analysis of measurements to be approximately unity[36]. Global chemical transport models often predict a much smaller relative reaction rate of ISOPOO with NO compared to $HO_2$ over the Amazon (~0.2), which was attributed to their order of magnitude lower soil NO emissions compared to measurements[36]. However, our WRF-Chem simulations predict that the ratio of reactions rates of ISOPOO with NO to that with $HO_2$ is ~1.0 over the background Amazon (averaged across the inner model domain during the local afternoons 16–20 UTC of the simulated period), which increases confidence in the model's ability to simulate the variation of isoprene oxidation products over the Amazon.

**Enhancement of $NO_x$ and oxidants by the Manaus plume**. Figure 3 shows that Manaus emissions significantly increase $NO_x$ and oxidant levels within the Amazon compared to the background. Under polluted conditions, the model indicates that urban $NO_x$ emissions are more than an order of magnitude higher than soil NO emissions. When Manaus emissions are turned on (top panels in Fig. 3), WRF-Chem also simulates an order of magnitude higher OH radical concentrations and significantly enhanced ozone compared to when Manaus emissions are turned off (bottom panels in Fig. 3). Consistent with our simulations, NO measurements aboard the G-1 aircraft show that in-plume NO concentration (1.3 ppb) is more than an order of magnitude higher than that observed over the background (0.04 ppb) (average across all aircraft transects during the simulated period). The model also predicts that reaction rates of ISOPOO with NO within the Manaus plume greatly exceed that with $HO_2$ by a factor of 3. Thus, the model predicts that urban NO emissions greatly increase the oxidizing capacity of the atmosphere and shift the atmospheric oxidation cycle towards the formation of nitrogen compounds[46]. Model predictions of this increased oxidation capacity within the plume are consistent with an analysis of measurements at the T3 site, which indicated that urban $NO_x$ amplifies OH concentrations by ~250% compared to the background[47]. Similarly, measurements suggest that ozone is also enhanced by a factor of 1.5–3 in plume-affected locations compared to background on March 13 (Supplementary Table 4), while the model predicts somewhat higher enhancement (factor of 2–3). The sharp increase in ozone and other pollutants between plume and background was also reported in two other recent WRF-Chem modeling studies in the Amazon basin[41,48].

We attribute the large observed enhancement of biogenic SOA within the urban plume (Fig. 1b) over the Amazon to the increase in oxidants due to $NO_x$ emissions. In comparison to the Amazon,

most other regions of the Northern hemisphere have much higher $NO_x$ levels[49]. Smaller plume-to-background differences in $NO_x$ concentrations over the Northern hemisphere could explain the smaller effects of $NO_x$ on biogenic SOA, reported previously[9].

Previous studies have also reported a larger sensitivity of SOA to POA that promotes condensation of semi-volatile SOA species[9,50]. In this study, we assumed that condensation of SOA is independent of POA. While this is a conservative assumption, simulated POA concentrations are much smaller compared to both biogenic and anthropogenic SOA (Supplementary Figure 5), so SOA formation in our simulations is not sensitive to this mixing assumption. Consistently, both ground-based and aircraft measurements using AMS have shown that the oxygenated organic aerosol (OOA) factor, which can be related to SOA, dominates over the primary organic aerosol factor (HOA) in the background Amazon[35,37].

**Biogenic SOA in the Amazon**. Figure 4 schematically illustrates how urban $NO_x$ emissions increase the reacted forest carbon (biogenic VOCs) over the Amazon, thereby enhancing biogenic SOA formation. In the absence of Manaus urban emissions, soil $NO_x$ emissions drive the oxidant cycling but lead to much lower SOA formation due to sub-ppb background $NO_x$ levels. Emission fluxes of biogenic VOC in the Amazon are modeled to be 80% isoprene, 17% monoterpenes, and 3% sesquiterpenes, on a mass

basis (Fig. 4a)[51]. Average daytime isoprene flux simulated by WRF-Chem (~5 mg m$^{-2}$ h$^{-1}$) agrees within 20% with average wet season isoprene emissions flux estimates (~6 mg m$^{-2}$ h$^{-1}$) derived from aircraft measurements using the Eddy Covariance technique, as reported by Gu et al.[52]. Note that although Gu et al.[52] found a strong correlation of isoprene emissions with terrain elevation during the dry season, the wet season did not exhibit this dependence. Since our study focuses on the wet season, the elevation dependence of isoprene emissions is not relevant to our study.

Simulated concentrations of isoprene and its first generation oxidation products (ISOPOOH, methacrolein and methyl vinyl ketone) agree with Proton Transfer Reaction Mass Spectrometer (PTR-MS) measurements aboard the aircraft within a factor of 2 (Supplementary Figure 3d). A comparison of model with aircraft measurements showed a factor of ~2 difference in monoterpene concentrations between model and measurements on March 13. However, both model and measurements show that monoterpene concentrations drop by a factor of 3 or more in the plume compared to the background levels due to enhanced in-plume oxidation of monoterpenes. It's noteworthy that monoterpenes measured by PTR-MS over the aircraft flight track were often close to their detection limit (~0.2 ppbv) on March 13. The model simulated background monoterpene concentrations (with a median simulated value of 0.6 ppb) presented in this study, however, are well within the range of other measurements over the Amazon (0.1–1 ppbv), as summarized by Alves et al.[53].

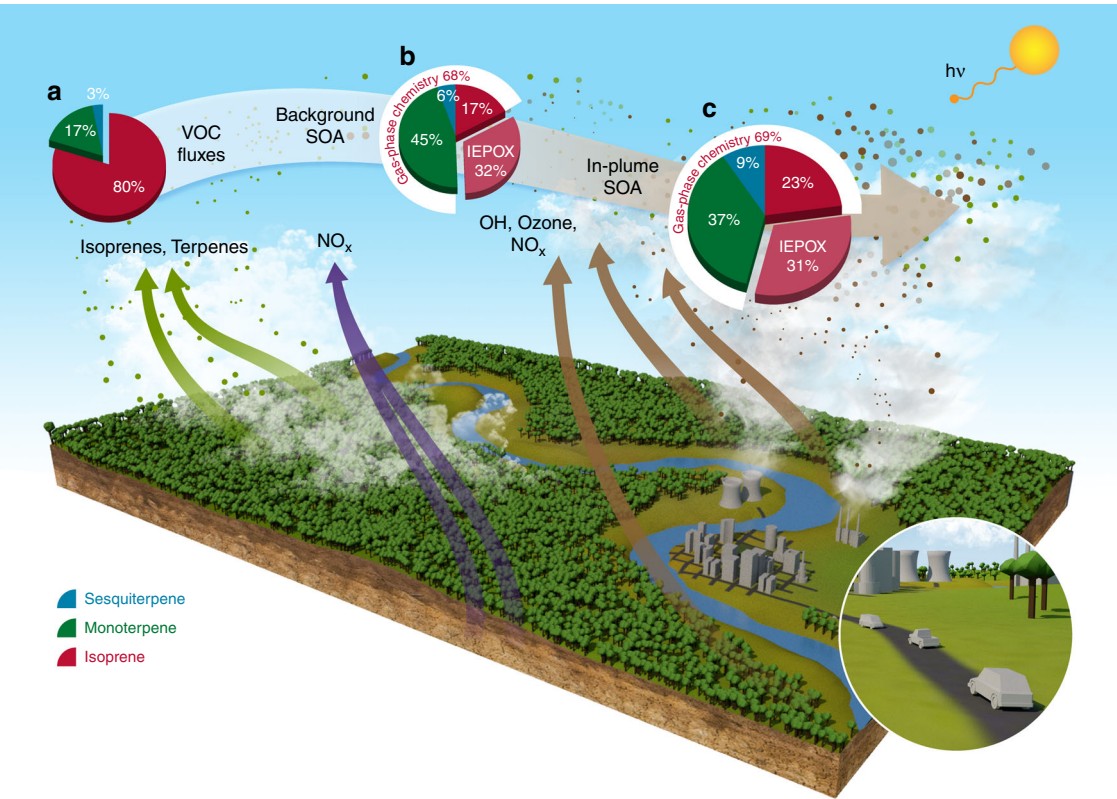

**Fig. 4** Schematic illustrating how $NO_x$ emissions from Manaus greatly enhance formation of biogenic SOA within the urban plume. $NO_x$ emitted by Manaus greatly increases oxidants (OH and ozone; brown arrows), which promote reaction of forest carbon (emitted as isoprene and terpenes; green arrows). In the absence of the urban plume, background soil $NO_x$ emissions (purple arrows) drive the oxidant cycling but are much smaller than the $NO_x$ emitted from Manaus. Lower background $NO_x$ causes smaller OH and ozone production, thus decreasing reacted biogenic VOCs and SOA formation. The pie charts indicate WRF-Chem simulated domain-averaged components of **a** Mass emissions fluxes of biogenic VOCs, **b** Background biogenic SOA and **c** In-plume biogenic SOA at 500 m altitude during the afternoon (16–20 UTC) of 13 March 2014. Biogenic SOA consists of two parts: gas-phase chemistry of isoprene, monoterpenes, and sesquiterpenes represented by VBS approach (~70% of total SOA), and multiphase chemistry that is driven by IEPOX uptake into SOA, as described in the text

Due to challenges in measurements and identification of sesquiterpenes, significant uncertainties remain about emissions fluxes of these species[54]. However, as discussed later in the manuscript, WRF-Chem simulated SOA from sesquiterpenes agrees with another observational study[55]. While the emission uncertainties of biogenic VOCs are considerable, and spatial heterogeneity will result in regional differences, the aircraft measurements demonstrate that our model simulation is a reasonable scenario for representing the biogenic VOC emissions in the Amazon region. Here, we use the WRF-Chem model to understand the contributions of different biogenic SOA precursors over the Amazon. The model predicts that ~70% of the total biogenic SOA is formed through the pure gas-phase chemistry pathway represented by VBS approach (Fig. 4b and c). All biogenic SOA types simulated by VBS are predicted to be enhanced in-plume. However, simulations indicate a greater in-plume enhancement of isoprene SOA (~180%) and sesquiterpene SOA (160%), compared to monoterpene SOA (~60%). The greater enhancement of isoprene and sesquiterpene SOA compared to monoterpene SOA can be explained by two additive effects of $NO_x$ on SOA formation: (1) $NO_x$ increases oxidants (OH and ozone), which increase the amount of reacted carbon for all biogenic VOCs, thus increasing SOA formation. (2) On a per reacted carbon basis, SOA yields may increase or decrease with $NO_x$ depending on specific VOC and oxidant types. For example, sesquiterpene photooxidation SOA yields increase while monoterpene SOA yields decrease as $NO_x$ increases[56]. This is most likely explained by the increased likelihood of sesquiterpenes oxidation to form multifunctional products of low volatility via isomerization, and also non-volatile organic nitrates at high $NO_x$ conditions compared to monoterpenes[56]. Despite the decrease of monoterpene SOA yields with increasing $NO_x$, the model predicts an enhancement of monoterpene SOA in plumes (~60%) due to the large competing effects through the increase of reacted monoterpenes through $NO_x$-promoted oxidant increase (effect 1 above). Consistently, a recent molecular level analysis of measured SOA over the southeastern United States showed that monoterpene SOA increases with increase in $NO_z$ (processed $NO_x$), even as the ratio of measured fragmentation to functionalization products increases during the daytime[8].

Isoprene SOA contributes ~50% to the total biogenic SOA (Fig. 4b, c) and comprises gas-phase oxidation and multi-phase IEPOX-SOA formation pathways. Model-predicted IEPOX-SOA contributions of ~30% (Methods) are within the range of previous analysis of measurements over the Amazon[23,25]. Under low $NO_x$ background conditions, isoprene photooxidation could contribute to SOA formation by pathways other than IEPOX uptake e.g. through the formation of low volatility dihydroxy dihydroperoxide, ISOP(OOH)$_2$[57]. We tested the potential of this pathway for isoprene SOA formation, but it was not found to be important over pristine Amazonia, due to a competing isomerization reaction of the peroxy radical precursor, which results in the formation of a higher volatility product[58] (reactions listed in Supplementary Table 5). Thus, in our approach, simulated isoprene SOA is the net effect of gas-phase chemistry pathways at high and low $NO_x$ conditions, and also multiphase IEPOX chemistry. Our results imply that under high $NO_x$ conditions within the Manaus plume, isoprene photooxidation SOA yields could be similar to or higher compared to background low $NO_x$ conditions[59,60]. Thus, effects (1) and (2) are also additive for isoprene, increasing isoprene SOA from gas-phase oxidation pathways in-plume.

Terpenes (sum of mono- and sesquiterpenes) together contribute the remaining 50% of the biogenic SOA (Fig. 4b, c). Although sesquiterpenes have much smaller emissions fluxes than monoterpenes and isoprene (Fig. 4a), they have higher yields.

Under background conditions, the model predicts a domain average sesquiterpene SOA contribution of 6%. This estimate is consistent with observations by Yee et al.[55] that sesquiterpene oxidation contributes at least 5% to total submicron OA mass in the background Amazon based on measurements of molecular tracers of sesquiterpene oxidation at T3 during the modeled period.

## Discussion

The vast Amazon rainforest transitions between pristine (pre-industrial like conditions) and urban-influenced polluted regions due to rapid developments with increasing electricity and transport demands and also deforestation for agricultural purposes[61]. This region provides a unique lens to investigate how chemical pathways of SOA formation have transitioned from preindustrial to urban-influenced present-day conditions. By combining analyses using a high-resolution regional model and laboratory, and field measurements during the GoAmazon2014/5 field campaign, we investigate how anthropogenic emissions enhance different pathways of biogenic SOA formation over the Amazon region. Both aircraft measurements and model predictions indicate ~60–200% average enhancements in OA concentrations with peak enhancements of ~400% in the Manaus plume compared to background regions. These SOA enhancements over the Amazon are much larger than those previously reported over more polluted regions like the continental United States[9,14]. A major factor contributing to this enhancement over the Amazon is the sharp transition in $NO_x$ from sub-ppb levels in the pristine background to more than several ppbs within the urban plume (~an order of magnitude increase), which greatly increases biogenic SOA. Simulations indicate that the large enhancement in biogenic SOA observed during GoAmazon2014/5 can mostly be explained by urban $NO_x$ emissions that increase oxidants, and no additional OH recycling mechanisms are needed to explain this enhancement. Our study also highlights that the contribution of terpenes to biogenic SOA formation is as important as isoprene even within the isoprene-dominated Amazonian forests. Our simulations demonstrate major shifts in biogenic VOC chemistry and SOA formation within locations affected by the urban plumes. On a per reacted carbon basis, VOC yields can increase or decrease as $NO_x$ increases due to complex VOC- and oxidant-dependent chemistry. However, the overall amount of reacted carbon increases through acceleration of oxidant cycling promoted by $NO_x$, thus increasing SOA formation over the Amazon. In addition, we show that although anthropogenic OA is a minor contributor to total OA over the Amazon, the major effects of urban pollution are manifested in terms of changing the chemical pathways and greatly increasing natural biogenic SOA formation over this region. Our results provide a clear picture of how anthropogenic emissions are likely to have greatly modified biogenic SOA formation since preindustrial times over the Earth, and imply that rapid urbanization in future years might substantially enhance biogenic SOA formation in the pristine forested regions of the Amazon.

## Methods

**Code availability**. We used the community regional Weather Research and Forecasting Model coupled to chemistry (WRF-Chem version 3.5.1[39,40]) for generating modeling results in this Manuscript. WRF-Chem is a community model and is accessible to users. Specific WRF-Chem configurations and modifications to gas- and particle-phase chemistry parameterizations used to generate results in this study are described below.

**WRF-Chem setup**. We use the regional WRF-Chem model[39,40] at cloud-, chemistry, and emissions-resolving scales i.e. at 2 km grid spacing, which is at a much higher resolution than that used in previous global modeling studies (typically ~100's of km)[62]. Since high-resolution simulations explicitly resolve features in

clouds, emissions, and chemistry, they do not suffer from uncertainties in parameterizations needed to represent these features in coarser resolution global models. Trace gases, aerosols, and clouds are simulated simultaneously with meteorology[40]. Biogenic VOC emissions are predicted using the Model of Emissions of Gases and Aerosols from Nature (MEGAN v 2.1)[51], which is coupled to the Community Land Model (CLM). CLM is run at the same grid spacing as WRF-Chem.

We use a nested grid configuration with an outer 10 km grid spacing domain covering $1500 \times 1000$ km and an inner 2 km grid spacing domain covering $450 \times 300$ km centered over Manaus City. Meteorological and chemical boundary conditions, land-surface scheme, and radiation scheme used for configuring the WRF-Chem runs used in this work are listed in Supplementary Table 1. The land surface data and emissions of trace gases and aerosols used for the simulations were the best available products for South America. The surface albedo, vegetation, and green fraction used in this study are documented in Beck et al.[63]. All model predictions analysed in this study are for the high-resolution inner domain that better resolves emissions, chemistry, and clouds compared to the outer domain. Also, the 2 km grid spacing inner domain explicitly resolves deep convective clouds, so no convective cloud parameterization is used for the inner domain.

The National Centers for Environmental Prediction (NCEP) Climate Forecast System Version 2 (CFSv2) reanalysis data (CFSR)[64] provides the meteorological initial and boundary conditions. Meteorological conditions were spun-up for 24 h, followed by 72 h of simulation, while the trace gas and aerosol species from the previous simulation were used as initial conditions. We conducted concatenated 4-day simulations, following the approach of Medeiros et al.[41] for this region. The chemical boundary conditions for trace gases and aerosols over the outer domain are provided by a quasi-global WRF-Chem simulation in 2014[65], while the inner domain received boundary conditions from the outer domain.

**Meteorological fields.** Supplementary Figure 6 shows that the model reasonably simulates the multi-day variations of several meteorological fields with measurements, including surface temperature, specific humidity, wind speeds, boundary layer height, downwelling solar radiation, and surface latent heat flux. The surface temperature, specific humidity, and wind speeds are averaged from 3 sites around Manaus and downwind areas (T1-Manaus, T2, and T3 sites). boundary layer height and downwelling solar radiation are taken from the T3 site, and surface latent heat flux is taken from the T0k site. The model is randomly sampled for 1000 grid points over land within 50 km radius centered at T3 and Manaus for all the meteorological fields except for latent heat flux. For latent heat flux, the model randomly sampled 200 grid points within 30 km radius of T0k site where latent heat flux above forest canopy from a tower measurement is available. A random sampling strategy to the model output is chosen to mimic large spatial variability from a few single point observations during the relatively short study period. Surface meteorology measurements at T3 are from ARM MET datastream[66], surface radiative flux measurements are from the ARM RADFLUX product[67]. Boundary layer height at the T3 site was derived using the vertical velocity statistics from the ARM DLPROFWSTATS4NEWS product[68]. The method follows Tucker et al.[69] by using profiles of Doppler Lidar measured vertical velocity variance as a measure of the turbulence within the boundary layer. Starting from the surface, the first vertical height level where the vertical velocity variance drops below $0.04 \text{ m}^2 \text{ s}^{-2}$ is designated as the boundary layer height.

**Emissions of trace gases and aerosols.** Since our WRF-Chem simulations are conducted at high resolution, including emissions of trace gases and aerosols were challenging for the Amazon, since detailed high-resolution emission inventories are scarce for this region. We combine several emissions inventories from different sources to get reasonable estimates of trace gases and aerosol emissions.

We include primary emissions of gases such as CO, non-methane volatile organic compounds (NMVOC), sulfur dioxide ($SO_2$), ammonia ($NH_3$), and oxides of nitrogen ($NO_x$) and aerosols, including organic carbon (OC), black carbon (BC) and sulfate ($SO_4$) from anthropogenic and biomass burning sources. Emissions of aerosols and gases from the traffic sector were included from a detailed high resolution $2 \text{ km} \times 2 \text{ km}$ gridded emissions inventory developed for this region based on the methodology described in a previous study[70]. We also included emissions of CO, $NO_x$, $SO_2$, VOCs, and particulate OC, BC, and $SO_4$ from power plants over the Manaus region including a mix of fuel oil, diesel, and natural gas used in 2014 for electricity generation and emissions from a large oil refinery based on a recent study[41]. Emissions of CO, $NO_x$, $SO_2$, VOCs, and particulate OC, BC, and $SO_4$ from these point sources were included. Additional emissions $SO_2$ and $SO_4$ area emissions were also included based on VOCA (http://bio.cgrer.uiowa.edu/VOCA_emis/) and the Emissions Database for Global Atmospheric Research (EDGAR v4.1), respectively. $NH_3$ emissions from industry, energy, residential, and agriculture are from the Hemispheric Transport of Air Pollution (HTAP_v2.2) 2010 emissions inventory[71].

**Biogenic and biomass burning emissions.** We included biomass burning emissions including both gases and aerosols from the 2007 Fire Inventory from NCAR (FINN07)[72]. FINN07 particulate emissions include organic carbon (converted to OA using an OA/OC ratio of 1.4), black carbon, $PM_{2.5}$, and $PM_{10}$. NMVOC

emissions from both anthropogenic and biomass burning sources are speciated according to the SAPRC-99 mechanism.

We also include emissions of biogenic volatile organic compounds (BVOC). BVOC emissions are derived from the latest version of Model of Emissions of Gases and Aerosols from Nature (MEGAN v2.1) that has been recently coupled within the land surface scheme CLM4 (Community Land Model version 4.0) in WRF-Chem[73]. The 138 biogenic species from MEGAN are lumped into 3 biogenic VOC classes: isoprene (ISOP), terpenes (TERP), and sesquiterpenes (SESQ).

**Unspeciated organic emissions.** Unspeciated organic emissions are traditionally not included in emission inventories, but are important for anthropogenic SOA formation[74–76]. About 10–20% of total non-methane organic gas (NMOG) emissions are not routinely included in emissions inventories[74]. These unspeciated emissions have significant potential to form SOA since they are semi-volatile or intermediate volatility organics (SIVOCs). We represent all unspeciated NMOG emissions as an intermediate volatility species (i.e. $C^* = 10^4 \text{ µg m}^{-3}$) for biomass burning and fossil-fuel sources referred to as a gas-phase species, IV-POA (g). Emissions of IV-POA (g) are assumed to be 20% of the total non-methane organic gas (NMOG) emissions for both biomass burning and fossil-fuel sources based on unspeciated fraction of NMOG emissions reported in Jathar et al.[74]. In addition, in our model, we assume that 50% of the emitted POA evaporates instantaneously and contributes to IV-POA (g), consistent with Jathar et al.[74], while the remaining 50% is assumed to be non-volatile. This reduces the number of POA tracers that need to be advected in the model and increases computational efficiency since our focus is mainly on SOA formation. Oxidation of the evaporated POA also contributes to anthropogenic SOA formation, as described later.

**Effects of soil NO emissions.** We included sources of NO emissions from soils within WRF-Chem. Previous studies suggest soil NO emissions for tropical forests in the range 20–60 µg NO $\text{m}^{-2} \text{ h}^{-1}$ [77–79]. However, much of this NO reacts within the canopy with ozone and does not enter the above-canopy atmosphere. This in-canopy reduction of NO reduces the effective flux of NO in the above-canopy atmosphere by ~75%. We choose the upper bound of soil NO emissions and reduce it by 75% to obtain an effective NO emissions flux of 15 µg NO $\text{m}^{-2} \text{ h}^{-1}$ ($8.3 \times 10^9$ molecules $\text{cm}^{-2} \text{ s}^{-1}$). This value is close to the soil $NO_x$ emissions range suggested by field measurements over Amazon rainforests (1.2 to $7.0 \times 10^9$ molecules $\text{cm}^{-2} \text{ s}^{-1}$), as discussed by Liu et al.[36]. Under background Amazonian conditions, Liu et al.[36] suggested that the relative reaction rate of isoprene peroxy radicals (ISOPOO) with $HO_2$ to that with NO is approximately unity. Indeed, our WRF-Chem simulations show that the ratio of reactions rates of ISOPOO with NO to that with $HO_2$ is unity under background conditions. This increases confidence in the ability of the model to simulate the relative reaction rates of isoprene peroxy radicals. In contrast, a previous study using the global model GEOS-Chem predicted a much smaller relative reaction rate of ISOPOO with NO compared to $HO_2$, which was attributed to its order of magnitude lower soil $NO_x$ emissions compared to measurements[36].

**Background sources of sulfate in the Amazon.** In addition to soil $NO_x$ emissions, we also included emissions of dimethyl sulfide (DMS) of 0.8 ng $\text{m}^{-2} \text{ s}^{-1}$ from local soil and plant emissions within the Amazon rainforest based on a recent study[80]. DMS is also advected from the oceans within our modeling domain. Oxidation of DMS results in the formation of $SO_2$, which is a background sulfate source. However, model simulations indicate that local DMS emissions are a minor source of sulfate, while the Manaus plume is a major source, which affects both in-plume and background sulfate concentrations. Simulated background sulfate of ~0.1 µg $\text{m}^{-3}$ agrees with aircraft measurements (e.g. on 13 March). The model simulates the increasing trends of sulfate within plumes compared to the background (not shown). However, in-plume sulfate simulated by the model is a factor of 2 higher than the observed sulfate, which is within the expected uncertainties of sulfate emissions sources within the Amazon.

**Simulating SOA using the VBS approach.** Simulated SOA from pure gas-phase chemistry pathway is represented in the model using a volatility basis set (VBS) approach. The VBS approach represents multiple generations of oxidation of biogenic VOCs that include isoprene, monoterpene, and sesquiterpene compound classes, and anthropogenic and biomass burning precursors using a lumped set of compounds. Initial yields are determined by fitting environmental chamber measurements and generally vary with VOC, $NO_x$, and oxidants (Supplementary Table 2). We modified the VBS approach to include further aging of organics at longer-timescale aging beyond that observed in environmental chambers, as described later in this section.

**Simulating anthropogenic SOA from unspeciated NMOG emissions.** Oxidation of anthropogenic IV-POA (g) by OH radicals results in the formation of semi-volatile SOA species that can be represented by fitting environmental chamber measurements using a VBS approach. Semi-volatile SOA formation yields due to oxidation of anthropogenic IV-POA (g) emissions were assumed to be the same as those reported for on- and off-road diesel vehicle sources and biomass burning/

wood burning from Table S3 in Jathar et al.[74] as shown below:

$$IV - POA(g) + OH = 0.044\,SVOC_1 + 0.071\,SVOC_2$$
$$+ 0.41\,SVOC_3 + 0.30\,SVOC_4. \qquad (1)$$

$SVOC_1$, $SVOC_2$, $SVOC_3$, and $SVOC_4$ represent lumped VBS species with $C^*$ of 0.1, 1, 10 and 100 µg m$^{-3}$, respectively. These initial yields represent the first few generations of chemistry measured in chamber experiments. The sum of particle-phase concentrations of $SVOC_1$, $SVOC_2$, $SVOC_3$, and $SVOC_4$ comprises anthropogenic SOA (Supplementary Figure 5e) in our study.

**Simulating natural biogenic SOA.** Since, the simulations are for the wet season and the Amazon is low in OA (measured OA ~1–2 µg m$^{-3}$), we rely primarily on chamber studies[56,59,81–85], which measured SOA yields at low concentrations. The SOA yields used in the model are determined by fitting chamber measurements of SOA mass evolution (documented in Supplementary Table 2). We selected most available chamber studies that measured SOA yields at low concentrations so that they could represent conditions over the Amazon. To the extent that extremely low volatility organic compounds (ELVOC)[86] and low volatility organic compounds (LVOCs) are not lost to the walls in the chamber experiments, these yields implicitly include the lower volatility compounds. Note that the key here is choice of measurements that measured SOA yields at low OA loadings. For example, using measurements in a continuous flow chamber, Shilling et al.[85] found SOA yield of 0.09 when 1.9 ppbv of α-pinene reacted to produce OA loadings of 0.9 µg m$^{-3}$. Importantly, this yield (0.09) remained constant at smaller OA loadings and the yield curve had no inflection point towards null yield for OA loadings as small as 0.15 µg m$^{-3}$. This result indicates formation of products having vapor pressures below 0.15 µg m$^{-3}$. More recent studies found that formation of ELVOCs likely explains the observation of no inflection towards null yield for α-pinene ozonolysis SOA observed at smaller OA loadings[86]. Thus, yields of these lower volatility compounds are implicitly captured by the 4-product volatility basis set fits with $C^*$ of 0.1, 1, 10, and 100 µg m$^{-3}$ applied in this study. However, because it is difficult to run chamber experiments at very small SOA loading (<1 µg m$^{-3}$), the fits to the chamber data will be insensitive to the specific value of the lowest $C^*$ bin chosen in the fits, but will be sensitive only to the fact that one such bin is included. In other words, fits to a typical chamber experiments will not be capable of distinguishing between products in $C^*$ bin of 0.1 or 0.01. For this reason most chamber fits choose a lower bound on the $C^*$ bin of 0.1, which also effectively captures mass in lower $C^*$ bins. This is just an inherent limitation of the laboratory experiments and the yield parameterizations.

Yields vary based on precursor type, oxidants (OH, ozone or nitrate i.e. $NO_3$ radicals) and also $NO_x$ levels during the measurements. The overall $NO_x$-dependent yield is calculated as a sum of high and low $NO_x$ yields weighted by $NO_x$ branching ratio[87] at each model grid point and time.

We include additional reactions for the VBS bins within the SAPRC-99 mechanism:

$$BVOC(g) + OH\,(or\,ozone,\,nitrate\,radical) \rightarrow \sum_{i=1}^{4} a_i BVSOA(g)_i, \qquad (2)$$

$$a_i = a_{i,high}B + a_{i,low}(1 - B), \qquad (3)$$

where BVOC(g) are the primary biogenic gas species (isoprene, terpene, or sesquiterpene), BVSOA(g)$_i$ represents SOA precursor species formed after photochemical oxidation of the BVOC(g), 'i' is the volatility bin (i = 1, …, 4 corresponding to $C^* = 0.1, 1, 10,$ and 100 µg m$^{-3}$), $a_i$ is the overall $NO_x$-dependent molar yield calculated from eq. (2), $a_{i,high}$ and $a_{i,low}$ are the molar yields under high and low $NO_x$ conditions, respectively, as shown in Supplementary Table 2, and $B$ is the $NO_x$ branching ratio as defined by Lane et al.[87]. In this work, we also included further $NO_x$-dependent multigenerational aging of both biogenic SOA and anthropogenic organics as described below.

**Further aging of VBS organics.** In the atmosphere, longer-timescale aging (beyond that observed in chambers) can change SOA yields compared to those determined from chamber measurements. Multigenerational aging results in both functionalization (decreasing volatility) and fragmentation (increasing volatility) reactions. In our previous studies[88,89], we showed that gas-phase fragmentation processes, which are often neglected in chemical transport SOA modeling parameterizations, could have large effects on both regional and global SOA loadings. In addition, the branching ratio between fragmentation and functionalization is reported to vary with the relative reaction rates between $NO_x$, $HO_2$, and $RO_2$ radicals. Gas-phase fragmentation is reportedly more prevalent under high-$NO_x$ compared to low $NO_x$ conditions[90,91]. In this study, we assume that the probability of fragmentation equals the branching ratio between peroxy-NO radicals reaction rates to the sum of all peroxy radical reactions rates (including peroxy-peroxy and peroxy-$NO_x$ reactions). However, we assign an upper limit of 75% fragmentation based on our previous sensitivity studies that varied this branching ratio (but without an explicit $NO_x$ dependence)[88,89]. Each generation of aging of the VBS

SOA species results in both functionalization and fragmentation reactions as a function of peroxy-$NO_x$ branching ratio, calculated at each WRF-Chem grid and time-step. In addition, we assume that a small fraction of organics fragment to species of much higher volatility and are not tracked. The maximum fraction of organics that is moved outside the VBS range is assumed as 10% by mass corresponding to the maximum fragmentation branching of 75%[88,89]. A sensitivity simulation, which turned off this additional aging showed a minor decrease in simulated mass concentrations of SOA in the background over the Amazon compared to the default simulation (not shown). We expect that the effect of $NO_x$-dependent multigenerational aging is less pronounced over the Amazon compared to more polluted locations (such as the continental United States) likely due to smaller background oxidant concentrations over the Amazon. Thus, the added multigenerational aging does not affect the main results and conclusions of this study.

**Aerosol treatments in MOSAIC module.** The condensation of low volatility gases ($H_2SO_4$ and $CH_3SO_3H$) and the dynamic partitioning of semi-volatile inorganic gases ($HNO_3$, $HCl$, and $NH_3$) to size-distributed liquid, mixed-phase, and solid atmospheric aerosols are represented by the Model For Simulating Aerosol Interaction and Chemistry (MOSAIC) aerosol module[43]. In this study, the aerosol species simulated in MOSAIC include sulfate, nitrate, ammonium, other inorganics (OIN), elemental carbon, organic carbon and aerosol water. We represented aerosols by 4-size sections with dry particle diameter ranges of 0.039–0.156, 0.156–0.624, 0.624–2.5, and 2.5–10.0 µm. Both interstitial and activated (cloud-borne) species corresponding to all aerosol chemical components are included and advected. Also, each simulated size bin includes both particle number and mass. The MOSAIC aerosol module includes treatments of nucleation, coagulation, and condensation as described in previous studies[43]. The size-dependent dry deposition of particles (both number and mass) is based on the approach of Zhang et al.[92]. In addition, both in-cloud and below-cloud wet removal of trace gases and aerosols are simulated following Easter et al.[93].

**Gas-phase chemistry.** Gas-phase chemistry in this study is based on the Statewide Air Pollution Research Center (SAPRC-99) mechanism[94], which includes 211 reactions of 56 gases and 18 free radicals. This mechanism is updated to include gas-phase photochemical oxidation of gas-phase organic species to form SOA particles. We include SOA formed due to oxidation of semi-volatile and intermediate volatility organic compounds (S/IVOC) emitted from anthropogenic and biomass burning sources (SI-SOA) and traditional SOA (V-SOA) formed due to oxidation of volatile organic compounds (VOC) precursors from biogenic emissions. We also extended this gas-phase chemistry mechanism to include isoprene epoxydiol (IEPOX) formation (Supplementary Table 5). VOC oxidation and catalytic effects of $NO_x$ on the oxidant cycle sustains the atmospheric oxidation capacity[95]. NO is necessary for $HO_x$ cycling and formation of ozone and OH radicals. Additional OH recycling mechanisms have been suggested in the literature[44], however, these recycling mechanisms often cause substantial overestimation of observed OH[45]. Therefore, in this study, we do not include additional OH recycling mechanisms in the model other than reactions between $HO_2$ and NO.

**Multi-phase IEPOX chemistry.** Multiphase SOA formation from isoprene oxidation is simulated using new aqueous chemistry modules that we added within WRF-Chem based on the simpleGAMMA model[42]. These aqueous chemistry modules are coupled to the model for simulating aerosol interactions and chemistry (MOSAIC), which simulates key inorganic species like sulfate, nitrate, ammonium ions, particle acidity, and water needed by the simpleGamma model[43]. The uptake of IEPOX within aqueous aerosols is determined by its solubility (Henry's law constant, $H_{IEPOX}$), followed by its reaction in the particle phase[42]. Here, we set $H_{IEPOX}$ as $1.7 \times 10^8$ M atm$^{-1}$ following Gaston et al.[24], which represents the higher end of $H_{IEPOX}$ values suggested in the literature[24,96–100]. Thus, IEPOX-SOA simulated in this study, most likely represents an upper bound estimate. Only a fraction of the epoxide reactively taken up by particles contributes to IEPOX-SOA formation[101]. The fraction of low volatility accretion products of IEPOX-SOA could vary significantly at different locations due to variable chemistry and partitioning. In this study, following measurements during GoAmazon2014/5 by Isaacman et al.[26], we constrained this fraction to 0.4 i.e. only 40% of IEPOX-SOA products persist in the particle-phase due to their low volatility in our simulations. Products of IEPOX reactive uptake that are semi-volatile evaporate from particles, leaving only low volatility accretion products as IEPOX-SOA and organosulfates[27].

**Key factors affecting computational cost of simulations.** Our simulations use detailed SOA parameterizations represented by the VBS approach, and a number of gas- and particle-phase VBS species need to be replicated for different source categories, including anthropogenic and biogenic classes (isoprene, terpene, sesquiterpenes classes) to resolve their individual contributions. Particle-phase species also multiply with number of size bins and also need to be replicated for interstitial and cloud-borne species that are advected in the model. Thus, a large number of gas- and particle-phase species (total of 420) are advected in the model, greatly increasing the computational cost compared to chemistry packages without SOA

within WRF-Chem. In addition, the high resolution nested grid configuration (2 km grid spacing) also increases WRF-Chem computational costs compared to global modeling studies that use much coarser grid spacings (~100–200 km grid spacings).

**Simulations with Manaus emissions on/off**. We compare WRF-Chem simulations with Manaus emissions on/off to quantify how Manaus emissions amplify oxidant cycling and biogenic SOA formation over the Amazon. Plume locations simulated by the model can be shifted compared to observations due to minor errors in simulated wind direction and dispersion. We conduct a careful analysis to identify the shifts in model-simulated plume compared to aircraft measurements. Figure 1 and Supplementary Figures 3 and 4 show that OA, ozone, CO, and $NO_y$ concentrations along measured and simulated flight transects can be used to accurately diagnose the shifts in simulated plume compared to measurements. The simulated CO baseline has some uncertainty depending on the boundary conditions (from global WRF simulations) and was adjusted by a constant value of ~30 ppb for better visual comparison with measurements. The key here is in-plume CO values are substantially larger than the background. Over the Amazon, $NO_y$ sharply increases within urban plumes by more than an order of magnitude compared to background locations and is used to identify the shifted plume in the model compared to observations. Our analysis in this study focuses on aircraft transects ~500 m altitude since they are within the mixed boundary layer during the daytime. Aircraft measurements represent a snapshot of changes that occur in biogenic SOA due to anthropogenic emissions over the otherwise pristine wet-season Amazon.

## Data availability

All data analyzed during the current study are included in this published article and its Supplementary Information. Aircraft measurements during the GoAmazon2014/5 field campaign used in this study are publicly available on the Atmospheric Radiation Measurement (ARM) website: http://campaign.arm.gov/goamazon2014/observations/. Model outputs from WRF-Chem that are used to generate figures in this study are available from the corresponding author on reasonable request.

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

## Acknowledgements

This work was supported by the U.S. Department of Energy (DOE), Office of Science, Biological, and Environmental Research's Atmospheric System Research (ASR) program. Dr. Shrivastava was also supported by the U.S. DOE, Office of Science, Office of Biological and Environmental Research through the Early Career Research Program. The authors thank the G-1 flight and ground crews for supporting the GoAmazon 2014/5 mission. Funding for data collection onboard the G-1 aircraft and at the ground sites was provided by the Atmospheric Radiation Measurement (ARM) Climate Research Facility, a U.S. Department of Energy Office of Science user facility sponsored by the Office of Biological and Environmental Research. The Pacific Northwest National Laboratory is operated for DOE by Battelle Memorial Institute under contract DE-AC06-76RL01830. R.Y. support at PNNL was provided by the US Department of Energy under the GoAmazon2014/5 project (Proc. no. 13/50521-7). J.A.T. was supported through a grant from the U.S. Department of Energy Office of Science DE-SC0018221. We acknowledge the support from the Central Office of the Large Scale Biosphere-Atmosphere Experiment in Amazonia (LBA), the Instituto Nacional de Pesquisas da Amazonia (INPA), the Instituto Nacional de Pesquisas Espaciais (INPE), and the Universidade do Estado do Amazonas (UEA and FAPEAM/GOAMAZON). P.A. was supported by FAPESP grants 2013/05014-0 and 2017/17047-0. The work was conducted under licenses 001030/2012-4 and 001262/ 2012-2 of the Brazilian National Council for Scientific and Technological Development (CNPq). Computational resources for the simulations were provided by the PNNL Institutional Computing (PIC) facility and EMSL (a DOE Office of Science User Facility sponsored by the Office of Biological and Environmental Research located at PNNL).

## Author contributions

M.S., S.T.M. and A.Z. designed research, M.S., S.L., J.E.S., S.R.S., Z.F., J.C., R.Y., Y.L. and C.Z. processed data and performed analyses, and M.S., M.O.A., P.A., H.M.J.B., L.K.B., J.B., R.C.E., J.F., J.D.F., Z.F., J.D.F., M.G., A.H.G., E.G.A., H.G., D.G., A.G., S.H.J., S.K., S.T.M., V.F.M., A.M., S.S.S., J.E.S., R.A.F.S., J.A.T., G.I.V.W., L.Y., R.A.Z., A.Z. and C.Z. wrote the paper.

## Additional information

**Competing interests:** The authors declare no competing interests.

