## [Peer Review File · Nature Communications]

Reviewers' comments:

Reviewer #1 (Remarks to the Author):

To evaluate the interactions between different pollutants to form SOA is an interesting and important topic with considerable impact on model development for possible climate- and land-use related changes. Therefore it is certainly interesting for researchers in the field. The paper is also well written and is based on the appropriate literature.

I am a bit irritated from two aspects that I feel need to be considered for improvement. First, the simulations include a full (anthropogenic and biogenic) emission as well as reduced emission run (neglecting biogenic emissions). This feels incomplete as long as a second reduced emission run without anthropogenic emissions is missing. I wonder how many SOA can be build up from biogenic emissions (including NO_x) alone, in particular since the emission effect doesn't seem to be additive.

Second, while the impact of biogenic emissions is important, the description and discussion of the variability of biogenic emissions is very limited. It is indicated in the method section that the MEGAN model is used to produce isoprene monoterpenes and sesquiterpenes, but there is neither an analysis about the uncertainty in these emissions nor a discussion about the impact of the actual large range of emissions and the compound composition within monoterpenes and sesquiterpenes. The reference of Gu et al. (Nature Communications) is cited to indicate that the approximate magnitude of emissions is met – however the original issue in this paper, namely the dependence of emission composition to elevation, is not. Is this a relevant issue for this area?

A minor thing in the end of the discussion is that the results of the study may be overstretched, if they are used to assume a future urbanization will lead to large SOA increases. Urbanization is unlikely to be spread through the whole region equally so the impact of anthropogenic emissions should be subject to a saturation process, correct?

Reviewer #2 (Remarks to the Author):

In this study the authors investigate the effect of urban pollution plume of SOA formation in the Amazon. The authors combine laboratory results of the effects of oxides of nitrogen on the SOA formation with a chemical transport model in order to get insights of enhancement of biogenic SOA formation due to urban pollution. In general the manuscript is well written and the topic is of interest for the aerosol community. The use of chemical transport model can provide important mechanistic knowledge on the observed variation of organic aerosol concentrations and this combined to the on-flight measurements is a powerful method. However, I have a concern about the treatment of organics in the model and, as the organics are in key role in this study, the concern reflects to the conclusions. Therefore, I find that revision is required.

(1) Treatment of the biogenic organics in the model:

The treatment of the biogenic organics in the model is not described with enough detail. At the moment, it is not clear to what extend the model represents the current best knowledge of the mechanisms and to what extend it is tuned to these particular field observations. The Table S4 reports biogenic SOA yield for "first few generations of oxidation products of isoprene, monoterpenes, and sesquiterpenes". Here the C* values extend from 0.1 to 100 ug/m³. Why are the low-volatile compounds excluded? At least for monoterpenes it has been shown in several studies that there are much less volatile compounds than the range considered here. According to the supplementary text (L147-149) these compounds under-go further reactions which lead to functionalization or fragmentation, but it is unclear how this is done in the model. How do these

reactions change C^* ? Further according to the text, gas phase chemistry of biogenic organics was also calculated based on SAPRC-99 mechanism. How are the VBS bins treated in this method?

Also, for isoprene the authors make the assumption that 40% of IEPOX-SOA products persist in the particle phase and this assumption is based on the measurements at the same site. So is the model tuned to match the measurements?

(2) Differences between the model and measurement:

In figure 1c the Transect 4 shows an underestimation of the in-plume enhancement by the model. This seems to be because overestimation of the background OA concentrations, since the in-plume OA was overestimated. Could the authors speculate on what causes this? Is the biogenic SOA formation in general overestimated in the model? I find that this is important also from the point of view of the conclusion that biogenic SOA is dominant in the in-plume conditions.

We thank the reviewers for their constructive comments. We have addressed these comments below.

Reviewer #1 (Remarks to the Author):

To evaluate the interactions between different pollutants to form SOA is an interesting and important topic with considerable impact on model development for possible climate- and land-use related changes. Therefore it is certainly interesting for researchers in the field. The paper is also well written and is based on the appropriate literature.

I am a bit irritated from two aspects that I feel need to be considered for improvement.

(R1.1) First, the simulations include a full (anthropogenic and biogenic) emission as well as reduced emission run (neglecting biogenic emissions). This feels incomplete as long as a second reduced emission run without anthropogenic emissions is missing. I wonder how many SOA can be build up from biogenic emissions (including NO_x) alone, in particular since the emission effect doesn't seem to be additive.

There seems to be a misunderstanding. As described on page 7, our second sensitivity simulation is without anthropogenic emissions i.e. Manaus emissions turned off. This particular run requested by the reviewer, has already been completed and the result is represented by the green line in Figure 1a. It shows that in absence of anthropogenic emissions, OA is equal to background biogenic SOA levels ($\sim 0.5 \mu\text{g m}^{-3}$). We have clarified the revised Manuscript by adding the following text:

“To understand how the Manaus plume affects biogenic SOA formation, we compare results from two model simulations: (1) Default, wherein all emissions including those from Manaus urban region and biogenic emissions are on, (2) A sensitivity simulation wherein biogenic emissions from the forest are turned on, but anthropogenic Manaus emissions are turned off (including NO_x, SO₂, anthropogenic VOCs, and primary particulate emissions, e.g., POA, sulfate). **This simulation represents background concentrations of OA and trace gases over the Amazon.**”

We also added the following text in the Manuscript:

“The difference between our two model simulations represented by the blue and green lines in Fig. 1a represents the enhancement of OA due to anthropogenic emissions.”

We did not do a simulation neglecting biogenic emissions since the objective of this study is to understand how biogenic SOA is modulated by anthropogenic emissions.

(R1.2) Second, while the impact of biogenic emissions is important, the description and discussion of the variability of biogenic emissions is very limited. It is indicated in the method section that the MEGAN model is used to produce isoprene monoterpenes and sesquiterpenes, but there is neither an analysis about the uncertainty in these emissions nor a discussion about the impact of the actual large range of emissions and the compound composition within monoterpenes and sesquiterpenes. The reference of Gu et al. (Nature Communications) is cited to indicate that the approximate magnitude of emissions is met – however the original issue in

this paper, namely the dependence of emission composition to elevation, is not. Is this a relevant issue for this area?

We have acknowledged the variability and uncertainty of biogenic emissions and included further discussions and clarifications in our response below, as well as adding additional text to the Manuscript.

The reviewer's point about the elevation dependence of isoprene emissions described by Gu et al. ¹ is valid. However, Figure 3 in Gu et al. ¹ shows that elevation dependence of isoprene emissions is important for the dry season (Fig 3a in Gu et al. ¹) but not the wet season (Figure 3b in Gu et al. ¹). Since our study focuses on the wet season, the elevation dependence is not an issue for our study, hence it was ignored.

We have clarified this by including the following lines in the Manuscript:

“Note that although Gu et al. ¹ found a strong correlation of isoprene emissions with terrain elevation during the dry season, the wet season did not exhibit this dependence. Since our study focuses on the wet season, the elevation dependence of isoprene emissions is not relevant to our study.”

As shown in Supplemental Figure S3d, WRF-Chem simulated concentrations of isoprene and their oxidation products (blue line) tracked the measurements (orange line). The differences between model and measurements are within a factor of 2 for isoprene and its oxidation products reflecting the uncertainties in emissions and their spatial heterogeneities. In comparison, the background simulation (with anthropogenic emissions turned off) shows larger isoprene and isoprene oxidation product concentrations due to smaller oxidant concentrations in the absence of anthropogenic NO_x emissions in this sensitivity simulation (green line in Figure S3d).

Monoterpenes: A comparison of model simulated monoterpene concentrations with aircraft measurements shows a factor of ~2 difference on March 13 (see figure below). However, both model and measurements show that monoterpene concentrations drop by a factor of 3 or more in the plume compared to the background levels due to enhanced in-plume oxidation of monoterpenes. It's noteworthy that monoterpenes measured by PTR-MS over the aircraft flight track were often close to their detection limit (~0.2 ppbv) on March 13. The model simulated background monoterpene concentrations (with a median simulated value of 0.6 ppb) presented in this study, however, are well within the range of other measurements over the Amazon (0.1-1 ppbv), as summarized by Alves et al. ². The above discussion has now been added to the main text of the Manuscript in the section: Biogenic SOA in the Amazon.

Figure: Measured and model predicted monoterpene concentrations along aircraft flight transects at 500-m altitude on March 13. WRF-Chem simulated monoterpenes are divided by a factor of 2 for easier visual comparison with measurements. Both model and measurements show a substantial decrease of monoterpene concentrations (by a factor of 3 or more) inside the plume compared to background due to enhanced in-plume oxidation promoted by increased oxidant concentrations within the plume, as discussed in the Manuscript.

Sesquiterpenes: Due to challenges in the measurement and identification of sesquiterpenes, significant uncertainties remain about emissions fluxes of these species³. However, as discussed in the manuscript, WRF-Chem simulated sesquiterpenes-SOA agree with an observational study⁴. Under background conditions, the model predicts a domain average sesquiterpene SOA contribution of 6%. This estimate is consistent with observations by Yee et al.⁴ that sesquiterpene oxidation contributes at least 5% to total submicron OA mass in the background Amazon based on measurements of molecular tracers of sesquiterpene oxidation at T3 during the modeled period.

We have now acknowledged these uncertainties in the Manuscript by adding the following lines to the main text:

“While the emission uncertainties of biogenic VOCs are considerable, and spatial heterogeneity will result in regional differences, aircraft measurements demonstrate that our model simulation is a reasonable scenario for representing the biogenic VOC emissions in the Amazon region.”

Based on this, we expect these emission are appropriate for achieving our main objective of demonstrating how biogenic SOA formation is affected by anthropogenic emissions.

(R1.3) A minor thing in the end of the discussion is that the results of the study may be overstretched, if they are used to assume a future urbanization will lead to large SOA increases. Urbanization is unlikely to be spread through the whole region equally so the impact of anthropogenic emissions should be subject to a saturation process, correct?

This is a valid point. We have modified our final summary statement as:
“Rapid urbanization in future years might substantially enhance biogenic SOA formation in the *pristine forested regions* of the Amazon.”

This statement clarifies that substantial enhancements will be seen due to urbanization in the pristine forested regions of the Amazon. Further urbanization in other regions that are already urbanized may see different effects such as a saturation process, as mentioned by the reviewer.

Reviewer #2 (Remarks to the Author):

In this study the authors investigate the effect of urban pollution plume of SOA formation in the Amazon. The authors combine laboratory results of the effects of oxides of nitrogen on the SOA formation with a chemical transport model in order to get insights of enhancement of biogenic SOA formation due to urban pollution. In general the manuscript is well written and the topic is of interest for the aerosol community. The use of chemical transport model can provide important mechanistic knowledge on the observed variation of organic aerosol concentrations and this combined to the on-flight measurements is a powerful method. However, I have a concern about the treatment of organics in the model and, as the organics are in key role in this study, the concern reflects to the conclusions. Therefore, I find that revision is required.

(R2.1) Treatment of the biogenic organics in the model:

The treatment of the biogenic organics in the model is not described with enough detail. At the moment, it is not clear to what extent the model represents the current best knowledge of the mechanisms and to what extent it is tuned to these particular field observations. The Table S4 reports biogenic SOA yield for “first few generations of oxidation products of isoprene, monoterpenes, and sesquiterpenes”. Here the C^* values extend from 0.1 to 100 $\mu\text{g}/\text{m}^3$. Why are the low-volatile compounds excluded? At least for monoterpenes it has been shown in several studies that there are much less volatile compounds than the range considered here.

This is a good point. The SOA yields used in the model are determined by fitting chamber measurements of SOA mass evolution (documented in Table S4). To the extent that extremely low volatility organic compounds (ELVOC) and low volatility organic compounds (LVOCs) are not lost to the walls in the chamber experiments, these yields implicitly include the lower volatility compounds. Note that the key here is choice of measurements that measured SOA yields at low OA loadings. For example, using measurements in a continuous flow chamber, Shilling et al. ⁵ found SOA yield of 0.09 when 1.9 ppbv of α -pinene reacted to produce OA loadings of 0.9 $\mu\text{g}/\text{m}^3$. Importantly, this yield (0.09) remained constant at smaller OA loadings and the yield curve had no inflection point towards null yield for OA loadings as small as 0.15 $\mu\text{g}/\text{m}^3$. This result indicates formation of products having vapor pressures below 0.15 $\mu\text{g}/\text{m}^3$. More recent studies found that formation of ELVOCs can likely explain the observation of no inflection towards null yield for α -pinene ozonolysis SOA observed at smaller OA loadings ⁶. Thus, yields of these lower volatility compounds are implicitly captured by the 4-product volatility basis set fits with C^* of 0.1, 1, 10 and 100 $\mu\text{g}/\text{m}^3$ applied in this study. However, because it is difficult to run chamber experiments at very small SOA loading (<1 $\mu\text{g}/\text{m}^3$), the fits to the chamber data will be insensitive to the specific value of the lowest C^* bin chosen in the

fits, but will be sensitive only to the fact that one such bin is included. In other words, fits to a typical chamber experiments will not be capable of distinguishing between products in C* bin of 0.1 or 0.01. For this reason most chamber fits choose a lower bound on the C* bin of 0.1, which also effectively captures mass in lower C* bins. This is just an inherent limitation of the laboratory experiments and the yield parameterizations.

Explicitly modeling the formation of ELVOCs would require incorporation of a chemical mechanism, which would be beyond the scope of this work and would also require a very careful re-evaluation of the chamber yield data, for the reason given above.

To address the reviewer's comments, we have added the above description to the Supplemental Information of the revised Manuscript under the section: Biogenic SOA yields.

(R2.2) According to the supplementary text (L147-149) these compounds under-go further reactions which lead to functionalization or fragmentation, but it is unclear how this is done in the model. How do these reactions change C*?

The C* bins do not change in the model. But the mass distribution of organic vapors changes due to functionalization and fragmentation, as organics move across different C* bins due to gas-phase chemistry, as described previously⁷⁻⁹. In this study, we assumed that the branching ratio between functionalization and fragmentation for gas-phase organics is NO_x dependent, based on previous studies. Gas-phase fragmentation is reportedly more prevalent under high NO_x compared to low NO_x conditions^{10,11}. However, as described in the supplemental, a sensitivity simulation, which turned off this additional aging showed a minor decrease in simulated mass concentrations of SOA in the background over the Amazon compared to the default simulation (which includes further aging). We expect that the effect of NO_x-dependent multigenerational aging is less pronounced over the Amazon compared to more polluted locations (such as the continental United States) likely due to smaller background oxidant concentrations over the Amazon. Thus, the added multigenerational aging does not affect the main results and conclusions of this study.

(R2.3) Further according to the text, gas phase chemistry of biogenic organics was also calculated based on SAPRC-99 mechanism. How are the VBS bins treated in this method?

SAPRC-99 is a lumped mechanism which includes 211 reactions of 56 gases and 18 free radicals. We include additional reactions for the VBS bins within the SAPRC-99 mechanism:

where $BVOC(g)$ are the primary biogenic gas species (isoprene, terpene or sesquiterpene), $BVSOA(g)_i$ represents SOA precursor species formed after photochemical oxidation of the $BVOC(g)$, i is the volatility bin ($i=1, \dots, 4$ corresponding to $C^* = 0.1, 1, 10$ and $100 \mu\text{g m}^{-3}$), a_i is

the overall NO_x dependent molar yield calculated from equation (2), $a_{i,high}$ and $a_{i,low}$ are the molar yields under high and low NO_x conditions respectively as shown in Table S2, and B is the NO_x branching ratio as defined by Lane et al.¹²

Equations (1) and (2) and the associated description provided above has now been added to the Supplemental Information.

(R2.4) Also, for isoprene the authors make the assumption that 40% of IEPOX-SOA products persist in the particle phase and this assumption is based on the measurements at the same site. So is the model tuned to match the measurements?

Previous studies have shown that products of IEPOX reactive uptake that are semi-volatile evaporate from particles, leaving only low volatility accretion products as IEPOX-SOA, and sulfates¹³. The fraction of low volatility accretion products of IEPOX-SOA could vary significantly at different locations due to variable chemistry and partitioning. Here we used measurements made during GoAmazon2014/5 by Isaacman et al.¹⁴ to arrive at a measurement-based estimate, and constrained this fraction to 40% in our simulations. We did not vary this fraction in an attempt to tune the model.

(R2.5) Differences between the model and measurement:

In figure 1c the Transect 4 shows an underestimation of the in-plume enhancement by the model. This seems to be because overestimation of the background OA concentrations, since the in-plume OA was overestimated. Could the authors speculate on what causes this? Is the biogenic SOA formation in general overestimated in the model? I find that this is important also from the point of view of the conclusion that biogenic SOA is dominant in the in-plume conditions.

Thanks for this comment. We found that the model accurately captures OA background concentration ($\sim 0.5 \text{ ug/m}^3$) as shown in Supplemental Table S2, however, the apparent model-measurement discrepancy in background OA concentrations for transect 4 in our previous version related to how the background was calculated in that version.

Transect 4 is farthest from the urban center. Errors in the simulated plume location and its dispersion are greatest for transect 4 compared to transects 1-3, as observed from CO comparisons for this day (Supplemental Fig. S3c). As shown in our previous submitted version of Supplemental Table S2, the model predicts background OA concentrations of $\sim 0.5 \text{ ug/m}^3$ for transects 1-3, which is in excellent agreement with measured background OA (0.5 ug/m^3). But the previous calculation showed that the model overestimates background OA for transect 4. In our previous approach, we were calculating background in the model using the same locations at which the aircraft measurements indicated background conditions. However, a complicating issue is that model simulated errors in plume location and dispersion are greatest at points that are farther downwind, therefore, the modeled background was, at least partially, affected by plume dispersion effects, especially for transect 4. This is also clear looking at Figure 1a, where, the blue line does not drop to the background value of $\sim 0.5 \text{ ug/m}^3$ at ~ 4000 seconds (flight time on X-axis).

To address this comment, we revised our calculations and used the simulation where anthropogenic emissions are turned off (green line in Figure 1a) to derive an average model

simulated background on this day. This background remained fairly close to 0.5 ug/m³ for all 4 transects in agreement with measurements. In this version, we revised our calculated OA enhancements as the difference between blue and green lines in Figure 1b. Due to this revised calculation of background OA concentrations, the model now overestimates enhancements by ~80% for transect 4. However, simulated OA enhancements show excellent agreement with measured enhancements (within 15%) for transects 1, 2 and 3, as discussed in the Manuscript and depicted in the revised Figure 1b. Note that enhancements are affected by model-measurement differences corresponding to both in-plume and background OA concentrations.

Thus, model simulated background estimate of OA concentrations in general agrees well with measurements (and is not overestimated).

References

- 1 Gu, D. *et al.* Airborne observations reveal elevational gradient in tropical forest isoprene emissions. *Nature Communications* **8**, doi:10.1038/ncomms15541 (2017).
- 2 Alves, E. G. *et al.* Seasonality of isoprenoid emissions from a primary rainforest in central Amazonia. *Atmos. Chem. Phys.* **16**, 3903-3925, doi:10.5194/acp-16-3903-2016 (2016).
- 3 Kesselmeier, J., Guenther, A., Hoffmann, T., Piedade, M. T. & Warnke, J. in *Amazonia and Global Change* 183-206 (American Geophysical Union, 2013).
- 4 Yee, L. D. *et al.* Observations of sesquiterpenes and their oxidation products in central Amazonia during the wet and dry seasons. *Atmospheric Chemistry and Physics Discussions* doi:10.5194/acp-2018-191 (2018).
- 5 Shilling, J. E. *et al.* Particle mass yield in secondary organic aerosol formed by the dark ozonolysis of alpha-pinene. *Atmos. Chem. Phys.* **8**, 2073-2088 (2008).
- 6 Ehn, M. *et al.* A large source of low-volatility secondary organic aerosol. *Nature* **506**, 476+, doi:10.1038/nature13032 (2014).
- 7 Shrivastava, M. *et al.* Global transformation and fate of SOA: Implications of low-volatility SOA and gas-phase fragmentation reactions. *J. Geophys. Res.-Atmos.* **120**, 4169-4195, doi:10.1002/2014jd022563 (2015).
- 8 Shrivastava, M. *et al.* Implications of low volatility SOA and gas-phase fragmentation reactions on SOA loadings and their spatial and temporal evolution in the atmosphere. *J. Geophys. Res.-Atmos.* **118**, 3328-3342, doi:10.1002/jgrd.50160 (2013).
- 9 Shrivastava, M. *et al.* Sensitivity analysis of simulated SOA loadings using a variance-based statistical approach. *Journal of Advances in Modeling Earth Systems* **8**, 499-519, doi:10.1002/2015ms000554 (2016).
- 10 Xu, L., Kollman, M. S., Song, C., Shilling, J. E. & Ng, N. L. Effects of NO_x on the Volatility of Secondary Organic Aerosol from Isoprene Photooxidation. *Environ. Sci. Technol.* **48**, 2253-2262, doi:10.1021/es404842g (2014).
- 11 Loza, C. L. *et al.* Secondary organic aerosol yields of 12-carbon alkanes. *Atmos. Chem. Phys.* **14**, 1423-1439, doi:10.5194/acp-14-1423-2014 (2014).
- 12 Lane, T. E., Donahue, N. M. & Pandis, S. N. Effect of NO(x) on secondary organic aerosol concentrations. *Environ. Sci. Technol.* **42**, 6022-6027, doi:10.1021/es703225a (2008).

- 13 Lopez-Hilfiker, F. D. *et al.* Molecular Composition and Volatility of Organic Aerosol in the Southeastern U.S.: Implications for IEPOX Derived SOA. *Environ. Sci. Technol.* **50**, 2200-2209, doi:10.1021/acs.est.5b04769 (2016).
- 14 Isaacman-VanWertz, G. *et al.* Ambient Gas-Particle Partitioning of Tracers for Biogenic Oxidation. *Environmental Science & Technology*, doi: 10.1021/acs.est.6b01674, doi:10.1021/acs.est.6b01674 (2016).

REVIEWERS' COMMENTS:

Reviewer #1 (Remarks to the Author):

The authors have sufficiently clarified my questions.

Reviewer #2 (Remarks to the Author):

The authors have addressed all my questions/comments and modified the manuscript where needed. This has removed the concerns I had regarding the manuscript, particularly the treatment of organics in the model. I recommend the manuscript to be published.